# Fog Droplet Size Distribution and the Interaction between Fog Droplets and Fine Particles during Dense Fog in Tianjin, China

**Qing Liu [1,2,3], Bingui Wu [2,4,*], Zhaoyu Wang [1] and Tianyi Hao [2,4]**

[1]    Tianjin Weather Modification Office, Tianjin 300074, China; lq1988314@126.com (Q.L.);
       13752663535@wo.cn (Z.W.)
[2]    Tianjin Key laboratory of Marine Meteorology, Tianjin 300074, China; tianyi_hao@126.com
[3]    Key Laboratory for Cloud Physics of China Meteorological Administration, Beijing 100081, China
[4]    Tianjin Meteorology Bureau, Tianjin 300074, China
*    Correspondence: tjwbgtjwbg@126.com

**Abstract:**   From November 2016 to January 2017, there were large-scale dense fog processes in Tianjin area on the west coast of Bohai Bay, China, even strong dense fog with visibility less than 50 m occurred.   Based on the observation data of fog droplet spectrum monitor, visibility sensor, environmental particle monitoring equipment and meteorological automatic station, the characteristics of fog droplet size distribution and the interaction between the fog droplets and fine particles during dense fog events were analyzed.  The results show following characteristics: (1) The average concentration of fog droplets ($N_a$), the average liquid water content ($L_a$) and the maximum liquid water content ($L_{max}$) in the strong dense fog process are larger than those in the dense fog. The average spectrum of fog droplet size distribution conforms to Junge distribution, and they are all broad-spectrum fog with a spectrum width of about 45 µm.  The average spectrum is similar to the dense fog of heavily industrialized inland in the world. (2) The maximum of fog droplet diameter during the formation stage have a good indication for the outbreak of strong dense fog.  (3) The mass concentration of $PM_{2.5}$ ($C_{PM2.5}$) is ranged from 121–375 µg/m$^3$, and the interaction between fog droplets and fine particles is analyzed.  During the formation, development and maturity stages, fog process can scavenge atmospheric fine particles, and the scavenging efficiency of $PM_{2.5}$ is more remarkable than $PM_{10}$. When $C_{PM2.5}$ does not exceed 350 µg/m$^3$, the increase in the concentration of fine particles is conducive to the rapid growth of fog droplets and the sharp drop of visibility. However, when $C_{PM2.5}$ exceeds the critical value, the increase has a negative feedback effect on the development of the fog process.  More investigations and cases are necessary to fully assess the mechanisms related to the dense fog events in Tianjin area and further analysis will be done.

**Keywords:** fog droplet size distribution; fog; atmospheric fine particles; liquid water content; $PM_{2.5}$; fog droplet spectrum

---

## 1. Introduction

The evolution from fog to dense fog is very rapid and difficult to predict, the sudden low visibility phenomenon is very easy to lead to sudden safety accidents [1]. Fog droplet microphysical parameters, such as number concentration (N), liquid water content (L), diameter (D), fog droplet size distribution and atmospheric fine particle concentration directly affect visibility (Vis) in fog [2] and determine the formation and dissipation of fog.  Clarifying the distribution characteristics and the rules of microphysical parameter changes at various stages of the fog life cycle can help improve forecasts of dense fog and strong dense fog [3], and provide a theoretical basis for fog modification [4].

In the past 30 years, more and more attention had been paid to the study of the microphysical characteristics of dense fog, and continuous progress had been made in the development and application of observation instruments of fog microphysical parameters. From the early "optical-electrical particle counter" and "three-purpose droplet spectrometer" to the widely used "FM-100" and "FM-120" fog monitor [5,6], the understanding of the microphysical characteristics of fog had been deepened. The formation of dense fog was accompanied by the broadening of the fog droplet size distribution in Alaska, USA [7]. British scholars analyzed the droplet size changes in the process of dense fog in Cardington, Bedfordshire, and found that the average droplet diameter in the early stage of fog outbreak was about 15–20 μm, so the starting signal of dense fog explosion may be clarified from the perspective of microphysics [8]. To some extent, numerical model skill also depends on the adjustment of microphysical parameters scheme. Moroccan scholars improved the skill of coastal fogs by adjusting the microphysics scheme of Casablanca area based on the droplet size distribution analysis [9,10]. If models could predict the N and L at each time step using a detailed microphysics parameterization, visibility could be calculated for warm fog conditions [11]. Early studies on the microphysical parameters of dense fog showed obvious regional differences in China. The N decreased while the L increased in order in urban areas (such as Nanjing and Chongqing), mountainous areas (such as Nanling and Lushan) and coastal areas (such as Zhoushan and Bohe). Many studies proved the D and the L are the main factors leading to low visibility in fog [12]. The N had a negative correlation with Vis. However the L of droplets had a positive correlation with the average diameter ($D_a$) [13].

In addition, atmospheric fine particles have a complex interaction with the growth of dense fog droplets in recent studies. The increase in the concentration of hygroscopic atmospheric particles can promote the formation of fog droplets even in unsaturated conditions, and the thickness of the fog layer will also increase with the increase of the concentration of atmospheric particles [14]. The intensity of fog will decrease with the decrease of aerosol concentration [15]. The hygroscopicity of atmospheric particles has a high dependence on particle size distribution and chemical composition, which changes with different conditions of relative humidity. So the growth process of fog droplets could be changed by the atmospheric hygroscopic aerosol [16]. For inland cities, the fine particles number concentration ranged from 5000 to 15,000 $cm^{-3}$ during Paris fog event, and the number concentration exerted a higher influence on the fog microphysical characteristics than chemistry (solubility) did [17,18]. A large amount of aerosol particles could act as condensation nuclei to enhance the formation of fog droplets in North China Plain [19]. For coastal cities, most aerosols (include $PM_{2.5}$) were effectively scavenged by fog water, while sulfur species could not be easily and effectively scavenged near the industrialized coastal area in Japan [20,21]. Deposition of fog droplets was the most important process for the evolution of the size distribution of aerosols in San Joaquin Valley fog in California [22].

In view of the frequent dense fog weather in North China in recent years [23–25], which has seriously affected aviation takeoff and landing. In order to improve the ability of fog modification, Tianjin Weather Modification Office introduced fog monitor equipment and environmental particle monitoring equipment to carry out microphysical observation experiments in the winter dense fog process in Tianjin, China, for the first time. It is hoped that this experiment can clarify the characteristics of fog droplet generation and the interaction between fine particles and droplet growth during the dense fog in Tianjin in winter, and provide a basis for future fog modification.

## 2. Measurement and Data Processing

The observation site is located at the Tianjin atmospheric boundary layer observation station [26] (39°04′ N, 117°12′ E) from November 2016–January 2017, which is located in the central part of the North China Plain, with the Taihang Mountains to the west, the Yanshan Mountains to the north, and the Bohai Sea to the east, at a distance of 50 km from its west coast. The geographical location of the observation station, designated Tianjin (TJ), is shown in Figure 1. The characteristics of the fog droplet size spectrum from this observation station were compared with those from the observation stations in other areas of China, also shown in Figure 1. Specifically, these stations comprised: NJBJ

(32°12′ N, 118°42′ E) in the northern suburbs of Nanjing [27,28]; GDZJ (21°1′ N, 110°32′ E) in Zhanjiang, Guangdong Province [29–31]; BJ (39°29′ N, 115°58′ E) in Beijing [32].

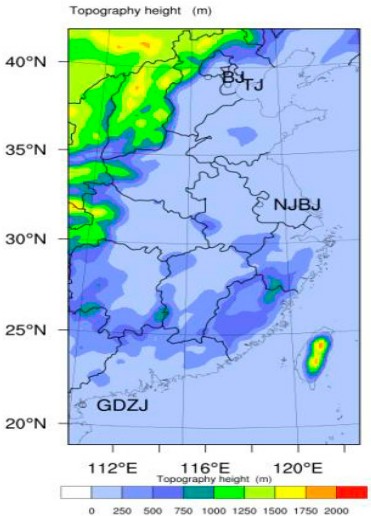

**Figure 1.** Tianjin test area and other mentioned fog test areas in China.

The data of fog droplet size distribution were obtained from an FM-120 fog monitor. The droplet diameter (D), liquid water content (L), number concentration (N) data were obtained with the trimmed-mean method [33]. This method follows this equation:

$$\overline{X_k} = \frac{X_{([n\overline{k}]+1)} + X_{([n\overline{k}]+2)} + \cdots + X_{(n-[n\overline{k}])}}{n - 2\,[nk]} \tag{1}$$

where n is the number of observations, k is a coefficient determined by people from 0 to 0.5 ($0 \le k < 0.5$), [ ] means to get integer, X represents sequential microphysical statistics. Because the amount of data is very large (time resolution is 1s) and the scientific value of the data must be guaranteed, we set k as 0.25 in this research. The visibility data were obtained from the MODEL 6000 forward scattering visibility meter, using minute-scale measurement data. The mass concentration of $PM_{2.5}/PM_{10}$ ($C_{PM2.5}/C_{PM10}$) were obtained from a TEOM (RP1405D) device, using hourly average mass concentration data. The wind speed data captured with the DZZ5 automatic weather station. The instrument specifications are listed in Table 1.

**Table 1.** Instrumentation and measurements for the winter fog observations in Tianjin.

| Instrument | Manufacturer | Model | Time Resolution | Measured Parameter |
|---|---|---|---|---|
| Fog monitor | DMT, US | FM-120 | 1 s | N, L, D, size distribution |
| Forward scatter visibility meter | Belfort, US | MODEL6000 | 1 min | Vis |
| Ambient particulate monitor | Thermo, US | TEOM (RP1405D) | 1 h | $C_{PM2.5}$, $C_{PM10}$ |
| Automatic weather station | Huayun Sounding, China | DZZ5 | 5 min | Wind, T, P, RH |

During the intensive measurement period, Tianjin experienced several fog events. Jointed to the weather phenomena and visibility data from the Meteorological Information Service System database of the China Meteorological Administration (CIMISS), every fog process was accompanied

by large-scale haze conditions, we called the weather phenomena as fog-haze processes. For all the fog-haze processes, individual (strong) dense fog event screening was performed based on the following three visibility levels: (1) Vis is not greater than 1500 m; (2) Vis first drops below 1000 m, then rises above 1000 m; (3) Vis stays less than 1000 m for more than 30 min. Six fog events were screened that met the above conditions.

Based on the grade of fog forecast in the national standard of the People's Republic of China (GB/T 27964-2011) [34], fog with Vis ≥ 50 m and < 500 m is classified as dense fog, fog with Vis < 50 m is classified as strong dense fog. Based on this classification, Case 1, 3, 4, 5, and 6 were dense fog events, and Case 2 was a strong dense fog event. Using the visibility as the criterion, the fog process can be divided into the four stages, the formation stage (when the Vis drops from 1500 m to 1000 m), the development stage (between the fog formation and the maturation stages), the maturation stage (after the Vis drops to its lowest level and remains basically constant for more than 30 min), and the dissipation stage (when the Vis increases noticeably from the lowest value to 1000 m).

Because this research mainly focuses on the changes of fog droplets and fine particles during the dense fog processes whose existing background is weak horizontal wind speed (Figure 2) and vertical wind [8], the changes of advection of fine particles could be ignored. The vertical transport may obviously affect the concentration of fog droplets and fine particles during the fog dissipation stage, while every continuous fog case was considered without any interruption, so the variation of fog spectrum originated from vertical movement is also neglected before fog dissipation. Moreover, being lack of vertical data of fog droplets and fine particles, following discussions about the characteristics and interactions of the droplet spectrum and fine particles at various stages of fog processes, were also based on the assumption that the variation of fog spectrum was only affected by local radiative cooling and the interaction of fine particles.

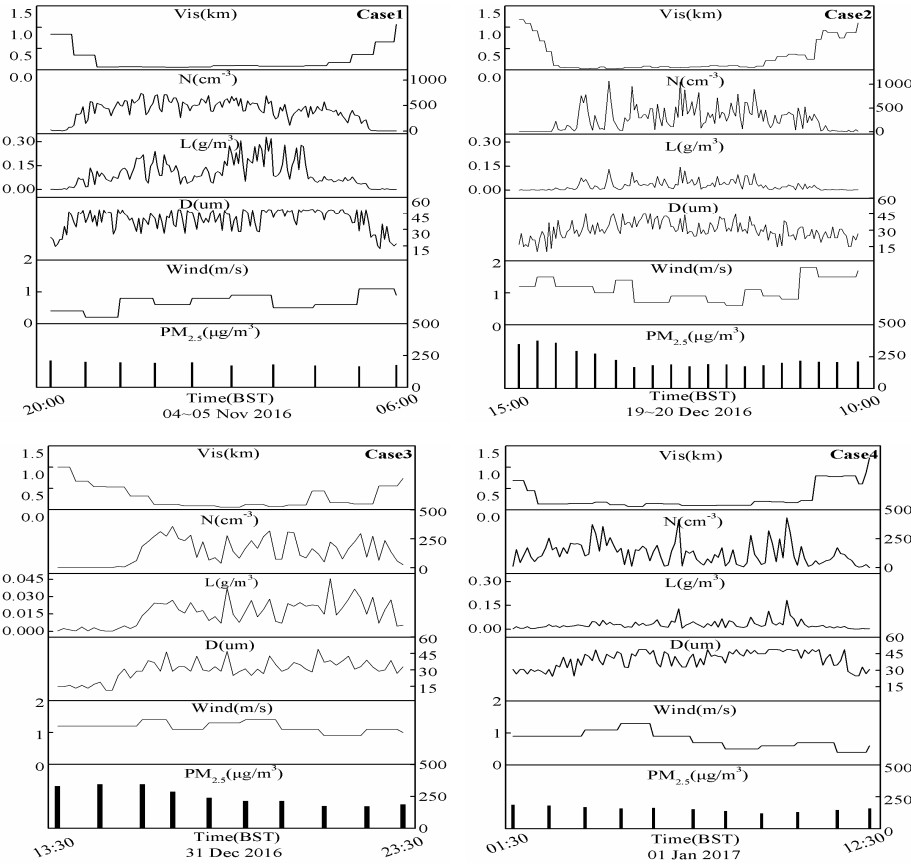

**Figure 2.** *Cont.*

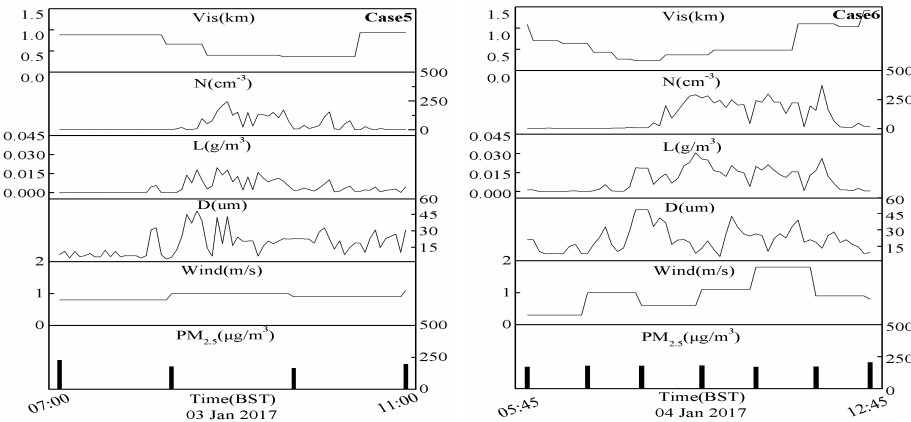

**Figure 2.** Temporal variations of 5-min averages of D, L, and N, and hourly averages of Vis, Wind speed and $C_{PM2.5}$ for the six fog processes.

## 3. Characteristics of Fog Droplet Size Distribution and Fine Particle

### 3.1. Evolution Characteristics of the Fog Droplet Size Distribution and Fine Particle

Table 2 lists the duration of Vis < 1000 m (Dr), the duration of the maturation stage ($Dr_{ma}$), the maximum/average concentration ($N_{max}/N_a$), the maximum/average liquid water content ($L_{max}/L_a$), and the maximum droplet diameter ($D_{max}$), as well as the range of visibility change ($Vis_{ma}$) during the maturation stage, and the maximum/minimum mass concentration of $PM_{2.5}$ ($C_{PM2.5max}/C_{PM2.5min}$). Figures 2 and 3 shows the change trend of D, L, N, Vis, Wind speed and $C_{PM2.5}$, we can find the following characteristics.

**Table 2.** Microphysical parameters and $C_{PM2.5}$ of the six fog processes in Tianjin.

| Case | Date | $Dr/Dr_{ma}$ (h) | $N_{max}/N_a$ (particles/ cm$^3$) | $L_{max}/L_a$ (g/m$^3$) | $D_{max}$ (μm) | $Vis_{ma}$ (m) | Classification of Fog | $C_{PM2.5max}$ (μg/m$^3$) | $C_{PM2.5min}$ (μg/m$^3$) |
|---|---|---|---|---|---|---|---|---|---|
| 1 | 04~05 Nov 2016 | 10/8 | 737/363 | 0.332/0.105 | 48 | 107~60 | Dense | 210 | 164 |
| 2 | 19~20 Dec 2016 | 19/10.5 | 1070/596 | 0.145/0.041 | 45 | 100~30 | Strong dense | 375 | 169 |
| 3 | 31 Dec 2016 | 10/3.5 | 356/172 | 0.046/0.018 | 49 | 120~70 | Dense | 346 | 173 |
| 4 | 01 Jan 2017 | 11/7.5 | 431/141 | 0.183/0.035 | 48 | 200~80 | Dense | 188 | 121 |
| 5 | 03 Jan 2017 | 4/0.5 | 247/102 | 0.020/0.010 | 49 | 400~370 | Dense | 230 | 167 |
| 6 | 04 Jan 2017 | 7/2.5 | 374/142 | 0.031/0.013 | 42 | 480~240 | Dense | 207 | 171 |

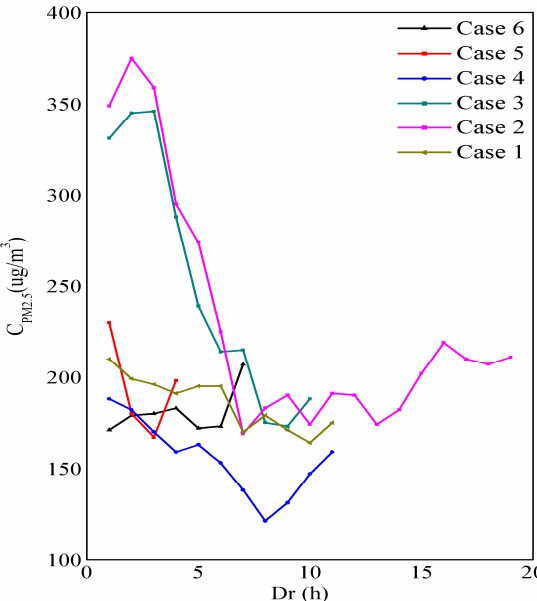

**Figure 3.** The temporal variations of $C_{PM2.5}$ for the six cases.

First of all, it can be seen from Figure 2 that the L, N, and D of Tianjin (strong) dense fog display good positive correlation relationships. This is consistent with the research results for Nanjing winter fog [27,28] and Zhanjiang coastal fog [29–31]. In terms of the spectral width characteristics of fog droplets, in the six fog processes, the spectral widths are at approximately 45 μm. It can be considered that the winter (strong) dense fog in Tianjin is a broad-spectrum fog based on suggestion of Niu et al. [35].

Furthermore, during the (strong) dense fog processes in Tianjin, $C_{PM2.5max}$ ranges from 188–375 μg/m$^3$, $C_{PM2.5min}$ ranges from 121–173 μg/m$^3$ (Table 2). Figure 3 and Table 2 reveal that $C_{PM2.5max}$ always appears in the formation stage (Case 1, 2, 3, 4 and 5) or the dissipation stage (Case 6). The trend of the decrease in the concentration of fine particles is basically consistent in each process, but the decline is slightly different, indicating that the fog droplets scavenge the fine particles during the development and maturation stages, and $C_{PM2.5}$ decreases significantly. Especially an "upturned tail" appears at the end of the temporal variations of hourly averages of $C_{PM2.5}$ in each process (Figure 3), which means the removal efficiency of fine particles in fog process is significantly reduced during the dissipation stage.

Over and above that, Table 2 reveals that the $N_a$, $L_a$, and $L_{max}$ of the strong dense fog (Case 2) are larger than those of dense fog. $N_a$, $L_a$, and $L_{max}$ of Case 2 are 596 particles/cm$^3$, 0.041 g/m$^3$, and 0.145 g/m$^3$, respectively, but they are 184 particles/cm$^3$, 0.036 g/m$^3$, and 0.122 g/m$^3$ in the dense fog cases, respectively. The calculation values of $N_a$, $L_a$, and $D_{max}$ of the (strong) dense fog cases in Tianjin, and the comparison of these values with the respective values of dense fog processes in several coastal and inland cities worldwide are listed Table 3. The $L_a$ in Tianjin is 0.037 g/m$^3$, which is very close to the $L_a$ in Nanjing [27,28], but it is significantly lower than that in Zhanjiang Coast [29–31], Nova Scotia [36,37] and Japan's Yodogawa Basin [20,38]. Moreover, Baoding's $L_a$ is the smallest [23,39]. Although Tianjin is also a coastal city, the $L_a$ during the (strong) dense fog process is more similar to the values found in inland areas. Comparing the main chemical composition of fog water and the concentrations of the main aerosols with several coastal and inland cities around the world (Table 3), we found that $C_{PM2.5}$ or $C_{PM1}$ are about 200 μg/m$^3$ in heavily industrialized inland area (Nanjing [27,28], Baoding [23,39], Kanpur [40,41]), as well as that in Tianjin. However, the concentrations of the main aerosols in several coastal areas (Nova Scotia [36,37], Yodogawa Basin [20,38], Fairbanks [42,43]) are much lower. The lower $L_a$ in Tianjin is due to the fact that under certain water vapor conditions, high $C_{PM2.5}$ will capture water vapor and form a large number of haze droplets, while high liquid water

content is mainly contributed by large fog droplets [44]. Moreover, the main chemical composition of fog water in Tianjin is similar to those of inland cities without obvious sea salt particles [45], which is significant difference from those of other typical coastal areas, maybe because of the observation spot being 50 km far away from Bohai Sea with the addition of its almost inland sea. Large-sized sea salt particles have strong hygroscopic growth capacity to form large fog droplets containing salt ions $Na^+$, and to supplement the liquid water content in the fog process. This may be the reason why $C_{PM2.5}$ is also large in Zhanjiang, $L_a$ is still greater than that in Tianjin.

**Table 3.** Microphysical characteristics of dense fog processes, the main chemical composition of fog water and the concentrations of the main aerosols in several coastal and inland cities around the world.

| Observation Site | $N_a$ (particles/ $cm^3$) | $L_a$ (g/m$^3$) | $D_{max}$ (μm) | Average Spectrum | Main Chemical Composition of Fog Water | Concentrations of Aerosols | Time |
|---|---|---|---|---|---|---|---|
| *Tianjin, China* | *253* | *0.037* | *47* | *Junge* | *Sulfate > particulate organic matter > elemental carbon > nitrate* [45] | $C_{PM2.5max}$ *188–375* μg/m$^3$, $C_{PM2.5min}$ *121–173* μg/m$^3$ | *2016 (This research)* |
| Nanjing, China [27,28] | 380 | 0.04 | 47 | Junge | Nitrate > Sulfate | 166 ± 96 μg/m$^3$ | 2007 |
| Kanpur, India [40,41] | 400 | - | - | Junge | Nitrate > Sulfate > Crustal elements | $C_{PM1}$ 199 μg/m$^3$ | 2012 |
| Baoding, China [23] | 350–500 | 0.001–0.01 | 50 | - | Sulfate | $N_{PM2.5}$ $10^4$ particles/cm$^3$ | 2011 |
| Yodogawa Basin, Japan [20,38] | - | 0.112 | - | - | Nitrate > Sulfate $Ca^{2+}$, $Na^+$, $Mg^{2+}$ | - | 2005 |
| Fairbanks, Alaska, USA [42,43] | 68 | - | 60 | Generalized gamma | Sulfur > Crustal compounds $Ca^{2+}$, $Na^+$, $K^+$ | 0.3 < D < 0.5μm 10 particles/cm$^3$ | 2012 |
| Zhanjiang Coast of Guangzhou, China [29–31] | 231 | 0.114 | 50 | Generalized gamma | Sulfate > Nitrate > Ammonium $Na^+$, $Cl^-$ | $C_{PM2.5}$ 100–200 μg/m$^3$ | 2011 |
| West coast of Casablanca, Morocco [9] | - | - | - | Generalized gamma | - | - | 2008 |
| Nova Scotia, Canada [36,37] | 78 | 0.092 | - | Generalized gamma | Sulfate> Nitrate > Organic matter $Na^+$, $Cl^-$ | $C_{PM2.5}$ 10 μg/m$^3$ | 1975 |

### 3.2. Statistical Characteristics of the Fog Droplet Size Distribution

Figure 4a shows the spectrum of fog droplet size distributions in the six cases. The spectrum exhibit exponentially decreasing trends. The distributions are skewed toward the small droplet side,

with the peak appearing in the small-droplet range of 5.4–7.2 μm in diameter. The peak diameter variation range of the droplets in the five dense fog processes is 5.4–6.1 μm, while the peak diameter in the strong dense fog process is 7.2 μm. These results indicate that the peak diameters of the spectrum are negatively correlated with visibility. The larger the peak diameter, the lower the visibility.

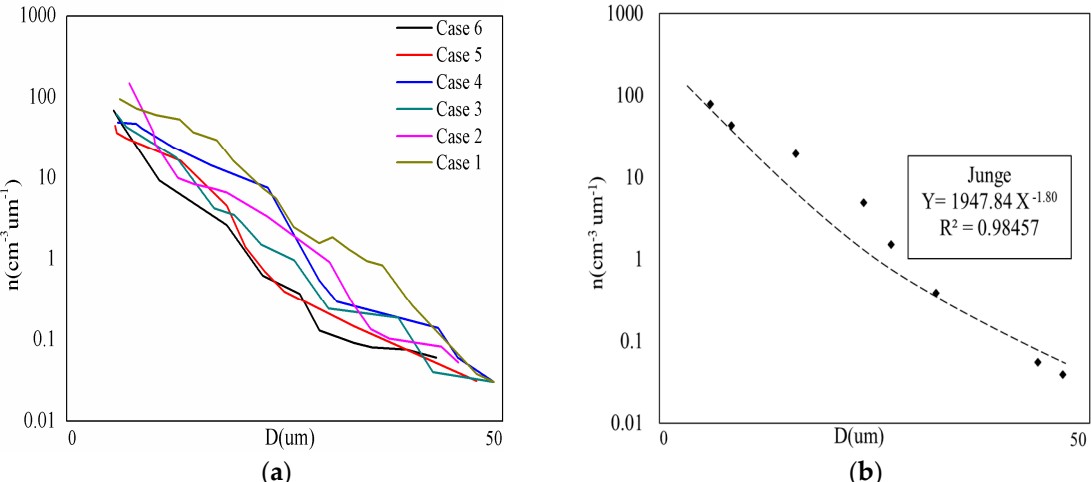

**Figure 4.** Fog droplet size distribution characteristics of six fog processes (**a**), and the average spectrum of fog droplet size distribution (**b**).

Researchers have demonstrated that the average spectral shape of fog droplet size distributions generally obeys either the Junge distribution [40] or the generalized gamma distribution [9]. The fitting form of Junge distribution follows this equation [46]:

$$n(D) = aD^{-b} \tag{2}$$

where a is the shape parameter, and b is the inverse dimension parameter. The goodness-of-fit coefficients $R^2$ can be determined using the following equation:

$$R^2 = \frac{SSR}{SST} = \frac{\sum_{i=1}^{n}(\hat{n}_i - \overline{n})^2}{\sum_{i=1}^{n}(n_i - \overline{n})^2} \tag{3}$$

where SST is the total sum of squares, SSR is the sum of squares of regression, $n_i$ represents for the each observation value of N, $\hat{n}_i$ represents for the regression value of N, $\overline{n}$ represents the average of N. Figure 4a shows the results of fitting spectral shape of each case and Figure 4b for the average spectral shape of the whole six cases in Tianjin using the least squares method [47]. All six cases fit the Junge distribution very well. Only the average spectrum is given as following:

$$n(D) = 1947.84D^{-1.80} \tag{4}$$

With $R^2 = 0.985$. Chenjiaping of Chongqing, China, winter dense fog average spectrums also fit the Junge distribution [13], the values of a is 2590, b is 3.4. Although the average spectrum of Tianjin and Chongqing all fit the Junge distribution, the droplet spectrums are slightly different, which is determined by the difference of a and b. b determines the shape of the spectrum. When b is closer, a determines the increasing/decreasing range of n(D) with D. So the increasing/decreasing range of n(D) in Tianjin is less than that in Chongqing. The results show that the average spectrum of winter (strong) dense fog in Tianjin presents an exponentially decreasing distribution, which is consistent with the average spectral type of inland fog in some part of China (such as Chongqing).

In coastal areas such as Fairbanks [42,43], Nova Scotia [36,37], Zhanjiang coast [29–31], and the west coast of Casablanca [9], the average spectral shape mostly obey the generalized gamma distribution.

In inland areas such as Nanjing, and Kanpur, the droplet spectra obey the Junge distribution (Table 3). Tianjin is a large city with a high population density situated on the west coast of Bohai Bay in North China Plain. Because the Bohai Sea is an inland ocean and its sea area is small, so its impact on Tianjin is smaller than that of the other open ocean. Therefore, Tianjin is still a continental climate, which is a coastal city greatly affected by industrialization and urbanization. Using Table 3 to continue to analyze the difference of the concentrations and chemical composition of hygroscopic particles in heavily industrialized inland and typical coastal areas. For the dense fog in Tianjin, fine particles have greater impacts on visibility than coarse particles, and his special chemical composition of hygroscopic particles in Tianjin (sulfate > particulate organic matter > elemental carbon > nitrate) [45] is similar to that of heavily polluted inland cities. The main chemical composition of fog water (whether it contains salt particles or not) has influence on the average spectrum. To sum up, due to its specific geographic location and chemical composition of hygroscopic particles, the fog droplet size distribution in Tianjin is similar to the distributions of heavily polluted inland cities but different from those of most coastal areas around the world.

It is envisaged that if the similarities and differences of the microphysical characteristics of fog droplets in the formation stage of dense fog and strong dense fog can be clarified, it may be possible to judge the strong dense fog outbreak in advance, thereby improving the ability to predict the strong dense fog. Based on this idea, we had a comparative analysis of the characteristics of the fog droplet spectrum in the four stages of fog, and focused on the formation stage. Figure 5 shows the maximum and average particle diameters of the droplets during the formation stages of dense fog and strong dense fog. The average values of the droplet size in the six fog processes were similar, ranging from 4.02–5.20 μm. However, the maximum droplet size varies widely. Five dense fog processes droplet sizes were 13.45 μm, 14.91 μm, 14.97 μm, 11.48 μm, and 16.98 μm, respectively. In the strong dense fog process (Case 2), larger fog droplets appeared, exhibiting a maximum droplet size of 24.95 μm. Based on the maximum droplet size distribution during the formation stage, it was divided into two ranges (Figure 5), Range A ($0 < D_{max} \leq 16.98$ μm) and Range B ($D_{max} > 16.98$ μm), respectively.

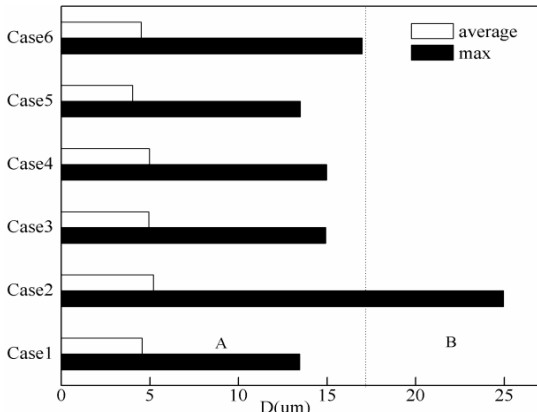

**Figure 5.** The maximum and average diameter during the formation stages of dense fog and strong dense fog.

Figure 6 shows the size distribution characteristics of fog droplets at each stage of the six (strong) dense fog processes. For the five dense fog processes, it can be seen that during the formation stage, the droplet size spectrum were narrow and the droplet diameters were small, with an average value of 4.50 μm. Thus, the air contained mostly haze particles or small-scale fog droplets, as well as a few large droplets (indicated by the stars in the Figure 6), with N values ranging from 1–3 per cm$^3$. The spectrum of condensation nucleus determines droplet concentration [48,49]. During the development process, the number of large particles that can act as droplet condensation nuclei increases sharply [19], the visibility quickly drops below 500 m, and the spectrum rises and noticeably widens. During the maturation stage, the number of fog droplets of all sizes reached their maximum

values for the process. The range of spectrum width was 42.4–48.9 μm, which became significantly wider. The finding that N did not decrease but rather increased during this stage is different from the phenomenon that large droplets increase and small droplets decrease due to collision-coalescence summarized by Liu et al. [27].

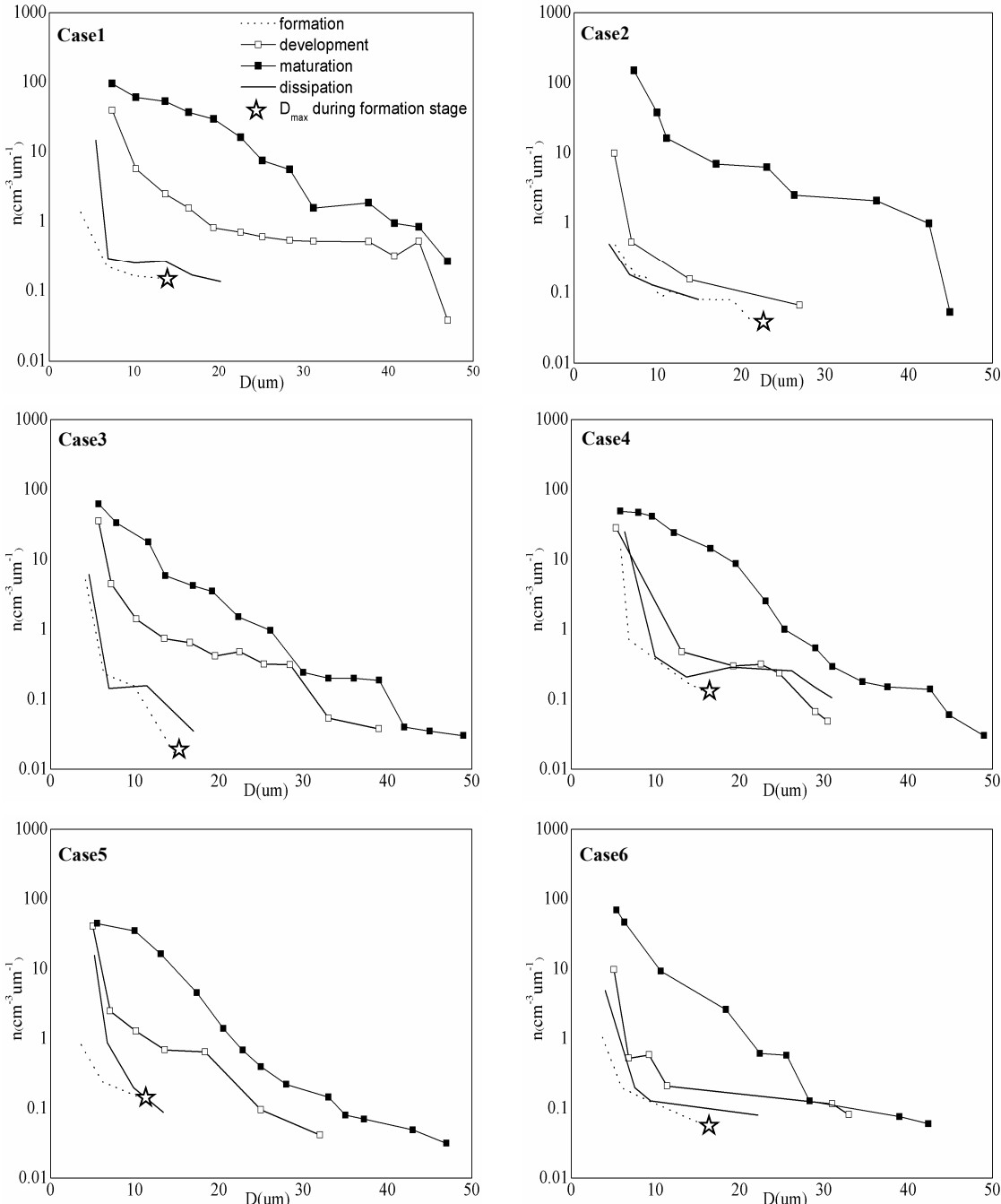

**Figure 6.** Fog droplet size distributions of (strong) dense fog in four stages and the $D_{max}$ during the formation stage. (The four phases are formation, development, maturation and dissipation. Case1, 3, 4, 5, 6 are dense fog. Case 2 is strong dense fog. The star in each plot denotes the size of the largest droplet and the corresponding N during the formation stage for the respective fog process).

For the strong dense fog, the spectral width during the formation phase is significantly wider than that during the dissipation phase (that is different from the five dense fog processes). A small number of fog droplet with value of 24.95 μm appear in short time (indicated by the stars in the Figure 6), which is

a good indication for the eruption of strong dense fog. During the development stage, the change trend of each microphysical parameter is consistent with the dense fog processes. During the maturation stage, the maximum particle size increased to 45 μm, the spectral width increased to 38 μm, the droplet concentration increased by two orders of magnitude, and $L_{max}$ suddenly increased to 0.105 g/m$^3$.

### 3.3. Interaction between Fog Droplets and Atmospheric Fine Particles

Select the strong dense fog process to further study the interaction between fog droplets and atmospheric fine particles. Case 2 was accompanied by severe haze, and the $C_{PM2.5max}$ and $C_{PM10max}$ were 375 μg/m$^3$ and 419 μg/m$^3$, respectively. The absolute changes of $C_{PM10}$ and $C_{PM2.5}$ in the four stages of fog are shown in Figure 7a,b. The maximum mass concentration of fine particles all appeared in formation stage of fog. During the stage of development and maturation, the mass concentration of fine particles show a rapid downward trend, which do not increase until the dissipation stage. Thus, the strong dense fog process in Tianjin has a strong effect on the removal of PM$_{10}$ and PM$_{2.5}$, which is consistent with previous studies in Tianjin [24]. The upward trend of the PM$_{10}$/PM$_{2.5}$, as shown in Figure 7c, obviously, after the formation stage, regardless of the median, 25% sample average, or 75% sample average, the ratio shows a steady upward trend, which fully indicates that the fog droplets are more significant clearance effect to PM$_{2.5}$ than PM$_{10}$. This conclusion is confirmed by the scavenging effect of water soluble aerosols by fog droplets during the dense fog in Indo Gangetic Plain [50].

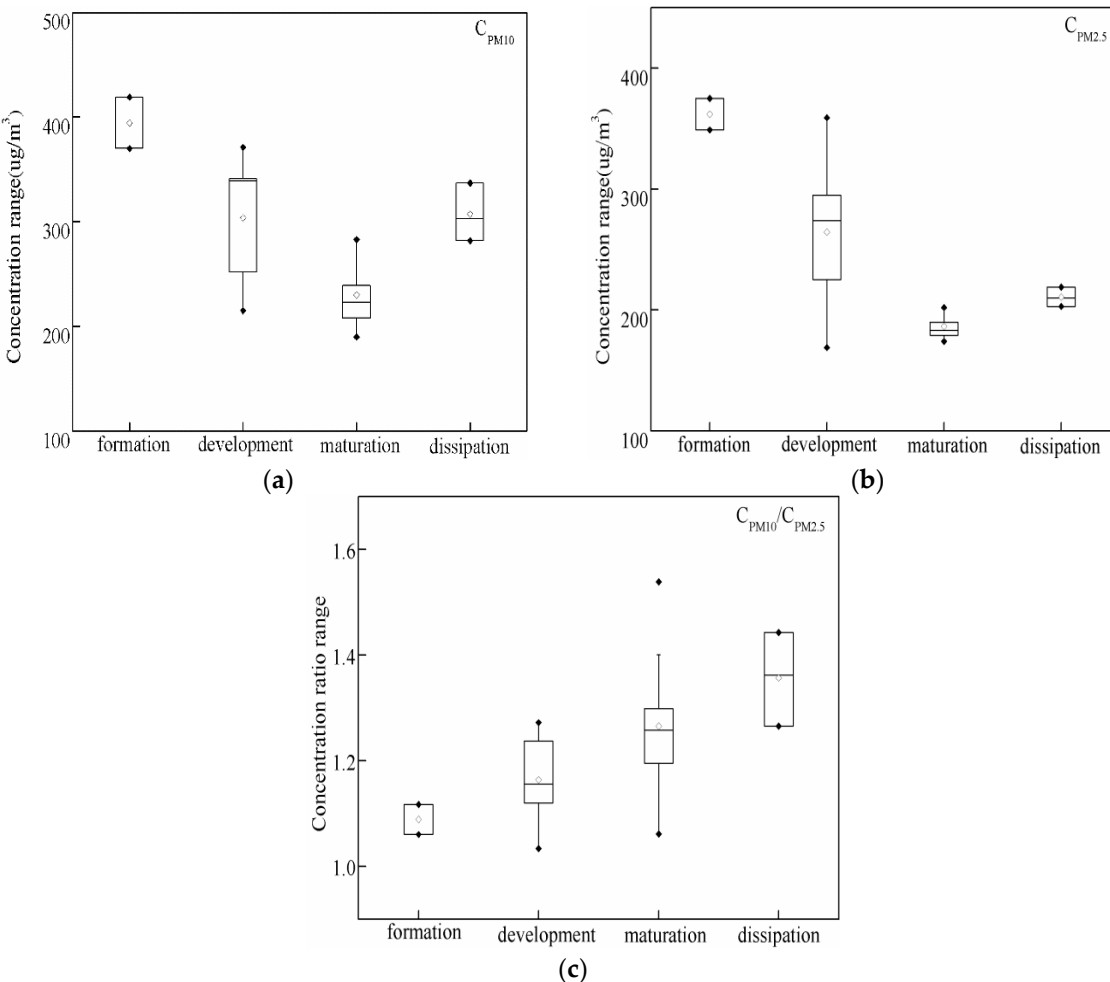

**Figure 7.** Changes of C$_{PM10}$(**a**), C$_{PM2.5}$(**b**), and C$_{PM10}$/C$_{PM2.5}$ (**c**)during the four stages in Case 2.

Figure 8 shows the distributions of L$_a$ and N/C$_{PM2.5}$ in different fine particle mass concentration intervals. For Case 2, Tian et al. [25] found that before fog appearance, the supply of exogenous water

vapor was sufficient, and specific humidity rised slowly. However, after fog appearance, there was no longer the transport of exogenous water vapor, and the local water vapor content decreases slightly compared with the early stage (figure is slightly). The data selected in this paper is after fog appearance, the water vapor content no longer changes significantly. Therefore, it can be approximated that the water vapor content is nearly constant. In the course of strong dense fog, with the increase of $C_{PM2.5}$, the value of $N/C_{PM2.5}$ and $L_a$ rapidly increased first. Especially when $C_{PM2.5}$ is in the range of 190–230 $\mu g/m^3$, the maximum value of $N/C_{PM2.5}$ appears. It is consistent with the conclusion of Yue et al. [29] that as the L increases, the ratio of the concentration of fog droplets to fine particles will increase. N increases with the rise of $C_{PM2.5}$, which indicates that when water vapor is sufficient, a large amount of $PM_{2.5}$ is activated as fog droplet condensation nucleus and $PM_{2.5}$ promotes the formation of fog. With the further increase of $C_{PM2.5}$ to 310 $\mu g/m^3$, $N/C_{PM2.5}$ remain stable, but $L_a$ value decrease rapidly. In other words, when $C_{PM2.5}$ is in the range of 230–310 $\mu g/m^3$, more $PM_{2.5}$ are simultaneously activated into fog droplets. However, with the increase of N, due to the limited amount of water vapor, a large number of fog droplets rob the water vapor, so that the water vapour content decreases rapidly. When $C_{PM2.5}$ is in the range of 310–350 $\mu g/m^3$, the value of $N/C_{PM2.5}$ rises again slightly, while the decrease trend of $L_a$ is unchanged. This shows that the competition of fine particles for water vapor is up to the peak period, and more $PM_{2.5}$ is activated as fog droplet condensation nucleus. As $C_{PM2.5}$ continues to increase in the range of 350–390 $\mu g/m^3$, the value of $N/C_{PM2.5}$ decreases instead, at the same time $L_a$ appears the minimum value. It is shown that excessive $PM_{2.5}$ no longer promote the increase of fog droplets, but instead consumes more water vapor. When the water vapor is not enough to support the activation of a large number of fine particles to form fog droplets, the condensed liquid water content will be lower ($L_a$ in the Figure 8).

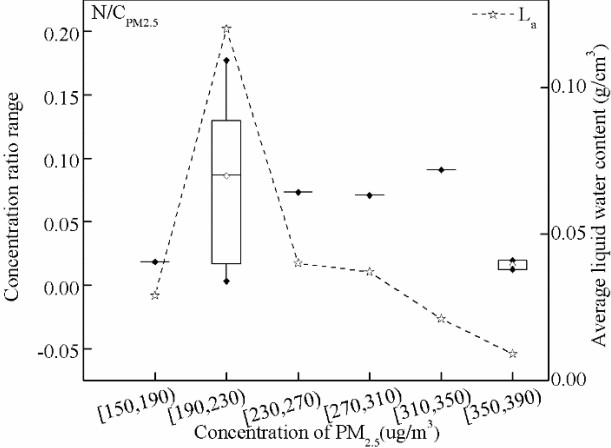

**Figure 8.** Changes of $N/C_{PM2.5}$ and average liquid water content with $C_{PM2.5}$ in Case 2.

Studies in Beijing, China have found that when $C_{PM2.5} > 200$ $\mu g/m^3$, the aerosol concentration has a strong effect on the rapid growth of dense fog droplets and the decrease in visibility [32]. The positive feedback effect on the growth of fog droplets becomes prominent when the $C_{PM2.5} > 230$ $\mu g/m^3$ in Tianjin (the $C_{PM2.5}$ value is higher than that in Beijing, China). Moreover, as described above, the interaction between $C_{PM2.5}$ and fog droplet growth in Tianjin is more complicated, because with the increase of $C_{PM2.5}$, its positive feedback effect on droplet growth will weaken, and even inhibit the growth of fog droplets. It is found by combining Figures 7b and 8, that during the development stage, the range of the $C_{PM2.5}$ value is the largest (215–371 $\mu g/m^3$), and only in the development stage, $C_{PM2.5}$ was found to be in the range of 310–350 $\mu g/m^3$. In other words, in the early stage of the development, the presence of a large number of fine particles is conducive to activation into more fog droplets, and also accelerates the condensation of liquid water content, and promotes the formation

and development of strong dense fog. However, at the later stage of the development, when the $C_{PM2.5}$ exceeds 350 μg/m$^3$, the growth of fog droplets is inhibited.

## 4. Conclusions

Considering the fog-haze harmful effect to the important port city and the largest open coastal city in North China, field observations of fog droplet size distribution and fine particles are set to improve the ability of fog modification at Tianjin atmospheric boundary layer observation station. Based on the analysis of microphysical parameters of fog droplet and fine particles from limited six (strong) dense fog cases, which were observed during November 2016 to January 2017. Some results about the chemical composition of fog water and aerosols in coastal and inland cities were cited to explain the observation phenomena, the summary is as following.

Due to its specific geographic location and background pollution, the fog droplet size distribution in haze days in Tianjin are fitted with Junge spectrum models, which is similar to the heavily industrialized inland areas. The spectral width of (strong) dense fog are approximately 45 μm and peak diameters ranges from 5.4–7.2 μm. The $D_{max}$ up to 16.98 μm during the formation stage have a good indication to the outbreak of strong dense fog. The interaction between fog droplets and fine particles during dense Fog in Tianjin is distinctive and bidirectional. On the one hand, the strong dense fog process have a strong effect on the removal of $PM_{10}$ and $PM_{2.5}$, and scavenging efficiency to $PM_{2.5}$ is more significant than $PM_{10}$. On the other hand, fine particles have different function to fog. Fine particles are benefit to enhance the fog process when $C_{PM2.5}$ is below the threshold, while showing contrary feedback effect and suppressing fog process when $C_{PM2.5}$ is larger than the threshold.

It should be noted, that the effect of dynamical parameters change (such as wind, humidity or turbulence) have not been neglected in fog processes, and the conclusions of this study are only drawn from the analysis of atmospheric microphysical characteristics. In addition, the threshold quantitative analysis of large droplets size during the fog formation stage and the spectral characteristic of fog droplets from only six (strong) dense fog events, more fog cases investigation need to enhance its credibility.

**Author Contributions:** Methodology: B.W., Q.L. and Z.W.; Instrumentation maintenance: Z.W.; Analysis of results: B.W., Q.L. and T.H.; Writing and figures: Q.L. All authors have read and agreed to the published version of the manuscript.

**Funding:** This work was jointly funded by the National Natural Science Foundation of China (41675018, 41675135, and 41705045), the Natural Science Foundation of Tianjin (17JCYBJC23400), the Bohai Rim Regional Fund (QYXM201801) and the Open Project of the Key Laboratory for Cloud Physics of China Meteorological Administration (2018Z01605).

**Acknowledgments:** We thank Zhang Hongsheng from Peking University for his helpful comments.

**Conflicts of Interest:** The authors declare no conflict of interest.

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
