# Peer review of "Fog Droplet Size Distribution and the Interaction between Fog Droplets and Fine Particles during Dense Fog in Tianjin, China"

_atmosphere, doi:10.3390/atmos11030258_

Round 1
Reviewer 1 Report
The manuscript presents analysis of measurements of several dense fog events in Tianjin, China. Observations were carried out using state-of-art instrumental complex that allows studying interactions between fog droplets and fine particles produced due to pollutions in detail. Both measurements and data analysis are technically solid, and the key conclusions reached in the paper seem to be fully supported by measurements. Despite some language issues, the manuscript is quite well written, easy to follow and reports some important conclusions regarding impacts of background pollutions on the formation and properties of dense fog that are certainly interesting to a wide readership of Atmosphere. The manuscript is overall of good quality, and can be published after a few issues are addressed.
Specific comments
I. Lines 18 and 330. The statement "Nucleation condensation is the main way of droplet growth in (strong) dense fog.' is obviously wrong and needs revision. First of all, there does not exist such a thing as 'Nucleation condensation' . Nucleation and condensation are two different processes. Nucleation is a New Particle Formation (NPF) from precursor molecules that has nothing in common with the condensational growth of nucleated particles.
II. Based on the conclusions made in the paper, background pollutions are the key factor controlling the formation of strong dense fogs. However, the concentrations and size distributions of hygroscopic particles in heavily industrialized inland and typical coastal areas should be quite different. Why this difference does not really matter?
III. While the authors claim that the main reason for the fog droplet distribution in Tianjin being similar to those in heavily polluted inland areas and different from those in typical coastal areas, no information is provided on the chemical composition and concentrations of gaseous precursors of nucleation/NPF. What were typical key pollutants in Tianjin area and their concentrations on the event days? Sulfuric acid, ammonia, amines, HOMs? Something else? Please, comment.
Technical issues:
While the paper is perfectly understandable, its language and style could be improved.
Reviewer 2 Report
In all sections, attention should be paid to clearly presenting the hypotheses, the working methodology, the data treatment, the representativeness of the results obtained in relation to the conclusions drawn, and the limitations. Moderate language on all these aspects is preferable, avoiding statements that are not scientifically proven.
More, the conclusion and abstract are not strongly supported by findings in the results section. Only 6 cases are not enough to do a proper characterisation of dense fog and strong dense fog events or to explore the correlation relationships. So, please analyse more cases in order to have proper conclusions.
For all calculated parameters please add the standard deviation and where is the case the uncertainties propagated through the calculus chain, and explain how the uncertainty was calculated.

Round 2
Reviewer 2 Report
The paper has been highly improved from the first version. Still the analysed data can not be considered sufficient to draw some conclusions. So please include, at your convenience in abstract or conclusion, a phrase stating that more investigations are necessary to fully assess the mechanisms related to the strong fog events in these area and further analysis will be done.
Also minor comments are listed below:
Line 28: I recommend the authors to replace the word “clarified”, with for example “analysed”
Line 118: Please add some reference, if existing, for the measurement site
Line 251: I recommend the authors to highlight in Table 3 the cases analysed in the paper
Line 484: I recommend the authors to replace the word “complicated”
